# Thromboelastometry-Guided Individualized Fibrinolytic Treatment for COVID-19-Associated Severe Coagulopathy Complicated by Portal Vein Thrombosis: A Case Report

**DOI:** 10.3390/biomedicines11092463

**Published:** 2023-09-05

**Authors:** Robin Forgács, Gergely Péter Bokrétás, Zoltán Monori, Zsolt Molnár, Zoltán Ruszkai

**Affiliations:** 1Department of Anesthesiology and Intensive Therapy, Flór Ferenc Hospital Kistarcsa, 2143 Kistarcsa, Hungary; forgacs.robin@florhosp.hu (R.F.); bokretas.gergely@florhosp.hu (G.P.B.); monori.zoltan@florhosp.hu (Z.M.); ruszkai.zoltan@semmelweis.hu (Z.R.); 2Department of Anesthesiology and Intensive Therapy, Semmelweis University, 1082 Budapest, Hungary; 3Centre for Translational Medicine, Semmelweis University, 1082 Budapest, Hungary; 4Department of Anaesthesiology and Intensive Therapy, Faculty of Medicine, Poznan University of Medical Sciences, 60-005 Poznan, Poland

**Keywords:** COVID-19, coagulopathy, fibrinolysis shutdown, viscoelastic testing, portal vein thrombosis, recombinant tissue plasminogen activator, systemic fibrinolysis

## Abstract

COVID-19-associated coagulopathy (CAC), mainly characterized by hypercoagulability leading to micro- and macrovascular thrombotic events due to the fibrinolysis shutdown phenomenon, is a life-threatening complication of severe SARS-CoV-2 infection. However, optimal criteria to assess patients with the highest risk for progression of severe CAC are still unclear. Bedside point-of-care viscoelastic testing (VET) appears to be a promising tool to recognize CAC, to support the appropriate therapeutic decisions, and to monitor the efficacy of the treatment. The ClotPro VET has the potential to reveal fibrinolysis resistance indicated by a clot lysis time (LT) > 300 s on the TPA-test. We present a case of severe SARS-CoV-2 infection complicated by CAC-resulting portal vein thrombosis (PVT) and subsequent liver failure despite therapeutic anticoagulation. Since fibrinolysis shutdown (LT > 755 s) caused PVT, we performed a targeted systemic fibrinolytic therapy. We monitored the efficacy of the treatment with repeated TPA assays every three hours, while the dose of recombinant plasminogen activator (rtPA) was adjusted until fibrinolysis shutdown completely resolved and portal vein patency was confirmed by an ultrasound examination. Our case report highlights the importance of VET-guided personalized therapeutic approach during the care of severely ill COVID-19 patients, in order to appropriately treat CAC.

## 1. Introduction

The COVID-19 pandemic caused by a positive sense single-stranded RNA genome virus—the Severe Acute Respiratory Syndrome Coronavirus 2 (SARS-CoV-2)—resulting in nearly 768 million confirmed cases with a tremendous mortality of about 7 million deaths worldwide, has presented an unprecedented challenge to the healthcare system.

Viral infection-associated immune response is characterized by the release of various inflammatory signal molecules (cytokines, interleukins, and other biomarkers), such as tumor necrosis factor alpha (TNF-α), interleukin-6 and 8 (IL-6, IL-8), C-reactive protein (CRP), and ferritin. As compared to other common respiratory viral infections, patients with COVID-19 have a higher risk of thrombotic events, indicated by elevations in serum D-dimer and fibrinogen levels, resulting in a more severe course of disease and poor outcome [1,2,3]. These findings are similar to the results of previous research during the H1N1 viral pneumonia outbreak, and Severe Acute Respiratory Distress Syndrome Coronavirus 1 (SARS), and Middle East Respiratory Syndrome (MERS) coronavirus epidemics [4,5].

SARS-CoV-2 may rapidly and unpredictably impair coagulation, leading to the so-called COVID-19-associated coagulopathy (CAC) characterized by micro- and macrovascular thromboembolic complications [6]. Measuring conventional parameters to evaluate patients’ coagulation system—prothrombin time (PT), activated partial thromboplastin time (aPTT), international normalized ratio (INR), fibrinogen, and D-dimer—is time consuming, hence they are not reliable enough to monitor these changes. There is convincing evidence, that point-of-care viscoelastic testing (VET)—such as thromboelastography (TEG) or rotational thromboelastometry (ROTEM)—can appropriately model each phase of coagulation ex vivo, thus it may immediately confirm coagulation disorders and even predict the risk of thromboembolic events or bleeding [7,8]. The most commonly used VET assays are the EX-, IN-, and FIB-tests. EX-test and IN-test assays model the extrinsic and intrinsic coagulation pathways, respectively, while the FIB-test containing platelet inhibitors (cytochalasin D and a synthetic GP2b3a antagonist) represents the tissue factor-induced activation of coagulation and the strength of fibrin polymerization. Viscoelastic traces can be divided into two parts. The first phase represents clotting (initiation, amplification, clot and fibrin formation), while the second phase provides information about fibrinolysis. In details, clotting time (CT) refers to the time in seconds needed for the first detectable clot formation; clot formation time (CFT) indicates the achievement of clot firmness in seconds; amplitude 5, 10, and 20 (A5, A10, A20) are the early parameters of clot quality measured at 5, 10, and 20 min timepoints after initiation, respectively; and maximal clot firmness (MCF) represents the maximal strength of the clot. Maximal lysis (ML) expressed as a percentage at the end of the measurement is a marker of fibrinolytic activity. It is an appropriate parameter in case of hyperfibrinolysis, but it only seems to be a surrogate marker when hypofibrinolysis is suspected. In summary, VET gives clear information mainly about the coagulation part of the hemostatic system; however, the novel ClotPro device (enicor GmbH, Munich, Germany; Haemonetics Corporation) also has the potential to evaluate the fibrinolysis shutdown phenomenon via the TPA assay containing 600 ng of recombinant tissue plasminogen activator (rtPA) that is added to 340 µL of whole citrated blood which will result in complete fibrinolysis indicated by normal lysis time (LT < 300 s), and 100% ML, if the fibrinolytic activity is preserved. In contrast, both prolonged LT and / or incomplete lysis (ML < 100%) indicate fibrinolysis resistance.

Several cofactors, inhibitors, and receptors regulate the fibrinolytic process. Plasmin, which is activated from plasminogen by two primary serine proteases—tissue plasminogen activator (tPA) and Urokinase-type plasminogen activator (uPA)—has a central role in fibrinolysis. Both activators have very short half-lives in circulation (4–8 min) due to the high concentration of their inhibitors (plasminogen activator inhibitor-1 and 2, PAI-1, PAI-2; α2-antiplasmin) [9].

Several pathological conditions may lead to impaired fibrinolysis (plasminogen activator/inhibitor ratio imbalances or decreased levels of plasminogen), resulting fibrinolysis resistance or the contrary, hyperfibrinolysis. Fibrinolysis resistance may play a central role in CAC [3], hence there is a strong rationale to monitor closely critically ill COVID-19 patients by point-of-care VET, in order to recognize and evaluate coagulation disorders as early as possible. This approach may have a crucial role in improving outcomes.

## 2. Case Description

We report a case of a 48-year-old male with a past medical history of paroxysmal atrial fibrillation without long-term oral anticoagulant treatment, and a recent history of diarrhea, generalized fatigue, shortness of breath, and cough lasting for 7 days prior to admission to the hospital. Positive result of a rapid antigen test from nasopharyngeal swab confirmed SARS-CoV-2 infection. On admission he was found in acute hypoxemic respiratory failure, with a peripheral oxygen saturation of 80%; however, it significantly improved after O_2_ supplementation via face mask. His blood pressure was 119/100 mmHg, he was tachyarrhythmic with a heart rate of 160/minutes, and atrial fibrillation was seen on ECG. Arterial blood gas analysis revealed metabolic acidosis (pH: 7.126, BE: −19 mmol/L), with markedly elevated lactate levels (12.6 mmol/L). Laboratory test confirmed elevated CRP, ferritin, transaminases, D-dimer, and troponin I levels indicating infection-associated multiple organ dysfunction (Figure 1). Thoracic CT scan confirmed extent bilateral ground glass opacities with severe consolidation in the inferior lobes of the lungs, while diffuse hepatic lesion without any signs of abnormalities in hepatic blood flow was found on abdominal CT scan.

Despite initial treatment—oxygen supplementation, volume resuscitation, antiarrhythmics (amiodarone, 300 mg, continued by 900 mg/24 h)—rapid deterioration developed and he was admitted to our Intensive Care Unit (ICU), where high flow nasal oxygen therapy and invasive hemodynamic monitoring (PiCCO, Getinge, Gothenburg, Sweden) were initiated in addition to standard treatment (remdesivir, dexamethasone, tocilizumab, stress ulcer prophylaxis). On admission to the ICU, patient’s APACHE II and SOFA scores were 18 and 9, respectively. Due to refractory atrial fibrillation ultra-short acting beta-blockers (landiolol), therapeutic anticoagulant treatment (enoxaparine, 2 × 80 mg) and antiplatelet therapy (acethylsalicilic acid, 300 mg) were also initiated. VET (ClotPro, enicor GmbH, Munich, Germany; Haemonetics Corporation) confirmed normal clot firmness, but prolonged clotting times (CT = 122 s, 177 s, 237 s, and 118 s on EX-, FIB-, IN-, and TPA-test assays, respectively) indicating an impaired initiation of clotting, while the fibrinolytic activity was mainly preserved. However, there was a signal of incipient fibrinolysis resistance on the TPA assay (LT = 325 s, ML = 95%).

After 24 h, severe hypoxemia and hemodynamic instability developed. His trachea was intubated, and mechanical ventilation was initiated. Due to refractory atrial fibrillation, a synchronous electrical cardioversion (2 × 200 J) was attempted, but after a short period of time, atrial fibrillation recurred. Due to high oxygen demand (fraction of inspired oxygen was 1.0), intermittent prone positioning was performed. Despite appropriate multimodal management, further deterioration occurred, serum levels of inflammatory markers increased, acute kidney injury developed, and a significant elevation in the levels of serum bilirubin and transaminases indicating acute liver failure were revealed.

Continuous renal replacement therapy (CRRT) was initiated for the developed acute renal failure in the form of continuous veno-venous hemodialysis (CVVHD) (OMNI, B. Braun Medical Inc., Melsungen, Germany) that was supplemented by adjunctive hemoadsorption (CytSorb, CytoSorbents Europe GmbH, Berlin, Germany) in order to treat hyperbilirubinemia and also the dysregulated, overwhelming host inflammatory response (Appendix A).

Conventional laboratory screening confirmed a significantly prolonged activated partial thromboplastin time (aPTT), a high international normalized ratio (INR), elevated D-dimer, and severely decreased serum fibrinogen levels (Figure 1). Prolonged CTs, normal clot formation times (CFT), and mean clot firmness (MCF) were seen on every VET assay (IN-test, EX-test, FIB-test, and TPA-test), while TPA-test indicated decreased fibrinolytic activity, then a few hours later a complete fibrinolysis shutdown was confirmed (Figure 2). Based on these findings, a portal vein thrombosis (PVT) was suspected, which was eventually diagnosed with an ultrasound scan.

Since prolonged CT on VET assay may be a result of ongoing anticoagulation therapy, or it can be caused by the low levels of clotting factors, 600 IU of prothrombin complex concentrate was given in order to normalize coagulation. On the other hand, fibrinolysis shutdown and subsequent portal vein thrombosis resulting in acute liver failure must have been treated. Several mechanisms contribute to the pathophysiology of the fibrinolysis shutdown phenomenon. Both the lack of tPA (absolute or relative due to early inactivation) and plasminogen may lead to fibrinolysis resistance. Furthermore, it has some pathophysiological rationale that the lacking amount of tPA needed to restore fibrinolytic activity can be counted from the patient’s estimated plasma volume and actual ML result of VET assay (plasma volume = blood volume x hematocrit; patient’s body weight: 95 kg, estimated plasma volume = 3460 mL; ML = 0%, lacking tPA ≈ 15 mg). Taking this knowledge into account, we gave an initial dose of 20 mg of rtPA (alteplase, 15 mg + 0.05 mg/kg first hour dose) continued by a rate of 15 mg/3 h (0.05 mg/kg/h), while 3 units of fresh frozen plasma (FFP) were administered, in order to substitute tPA and its substrate, plasminogen. We monitored the efficacy of the treatment with repeated VET assays every 3 h. Until fibrinolysis shutdown completely resolved, indicating satisfactory fibrinolytic therapy, we gave further boluses of rtPA and FFP, applying a total dose of 105 mg and 9 units, respectively (Figure 3).

After successful treatment, laboratory and oxygenation parameters indicated by the increasing PaO_2_/FiO_2_ ratio improved significantly (Table 1), while portal vein patency, and subsequent restoration of hepatic circulation were confirmed by a repeated ultrasound scan.

Our patient’s clinical condition improved further during the next 2 days, but unfortunately on the 6th day of treatment a refractory ventricular fibrillation occurred and he died. Autopsy revealed severe viral myocarditis as the cause of death.

Our patient’s clinical course, symptoms, and assigned interventions are summarized in Figure 4.

## 3. Discussion

COVID-19-associated coagulopathy, usually characterized by both hypercoagulant state and the fibrinolysis shutdown phenomenon, is common in critically ill COVID-19 patients. In May 2021, the National Institutes of Health (NIH) recommended CAC as a high-priority research topic [6]; since then, several molecular and cellular pathways have been identified.

SARS-CoV-2 infection may cause vascular endothelial cell dysfunction and damage to the glycocalyx leading to disruption of the antithrombotic functions of the endothelium [10]. High affinity of SARS-CoV-2 for the endothelial ACE2 cell surface receptors leads to ACE2 deficiency at the site of infection, resulting in a dysregulated kinin degradation and subsequent increase in vascular permeability. Furthermore, SARS-CoV-2 activates the complement system, enhances platelet activation, and induces the release of procoagulant factors, while the intrinsic anticoagulant activity becomes suppressed [10,11,12]. Wójcik and colleagues reported that COVID-19 patients may have significantly lower protein C activity, decreased free protein S levels, and ADAMTS13 antigens, but increased von Willebrand Factor (vWF) and a higher prevalence of antiphospholipid antibodies, while their thrombin generation potential was similar to controls [13]. Contribution of these dysregulated pathways drive CAC and may lead to immunothrombosis and multiple organ dysfunction. Considering this pathophysiology, combined antiplatelet-anticoagulant treatment was recommended in critically ill COVID-19 patients; however, this approach has not provided significant advantages in preventing CAC and subsequent organ failure [14,15]. Our case also highlights this problem and supports previous findings. Despite antiplatelet and anticoagulant therapy, a portal vein thrombosis developed indicating that standard medical treatment was insufficient to prevent macrovascular thrombotic complications.

A significant downregulation in uPA and tPA gene expressions, with an increased expression of genes encoding PAI-1, was found in SARS-CoV-infected mice, suggesting a pivotal role of fibrinolysis resistance in the course of CAC [16]. While it is obvious to substitute tPA in cases of life-threatening macrovascular thromboembolic events (e.g., pulmonary embolism, ischemic stroke), but considering fibrinolytic therapy in COVID-19-associated ARDS that likely resulted from a microvascular thrombosis in the lungs, also has some rationale. This hypothesis is supported by a case series of successful fibrinolytic treatments resulting in marked improvements in oxygenation in patients with COVID-19-associated ARDS [17]. Regarding oxygenation, we found similar effects after fibrinolytic therapy. Although we cannot state obvious benefits of the treatment on mortality, the improvement in a physiological endpoint—the PaO_2_/FiO_2_ ratio—is striking and thought-provoking, hence considering this outcome may be an important aspect in future research.

CAC is a dynamically changing disorder, hence conventional coagulation parameters (PT, aPTT, INR, D-dimer, and fibrinogen) are not reliable enough to monitor these changes. Point-of-care VET can appropriately model each phase of coagulation ex vivo; thus, it may immediately confirm impaired coagulation [7]. Viscoelastic traces are divided into the clot formation and the clot lysis phases of the coagulation process. The first part of each curve represents clotting, while the maximal lysis (ML) expressed as a percentage at the end of the measurement is an appropriate parameter in case of hyperfibrinolysis, but it only seems to be a surrogate marker when hypofibrinolysis is suspected [8,18]. The novel ClotPro system has the potential to evaluate the fibrinolysis shutdown phenomenon via the TPA assay containing 600 ng of rtPA. The TPA assay will result in complete fibrinolysis if the fibrinolytic activity is preserved; in contrast, both prolonged lysis time (LT > 300 s) and / or incomplete lysis (ML < 100%) indicate fibrinolysis resistance [19].

Almskog et al. found that ROTEM MCF values assessed at hospital admission were significantly higher in COVID-19 patients as compared to healthy controls, and higher in critically ill patients. They suggested that VET may help distinguish the level of care required [20]. Kruse et al. identified hypofibrinolysis as an important mechanism in CAC and suggested that thromboelastometry results evaluated together with serum D-dimer levels may help in identifying patients requiring anticoagulation [21]. Their findings were strengthened by a study conducted by Wright and colleagues who reported that combination of high D-dimer levels and hypofibrinolysis indicated by VET showed the best predictive value for thrombotic events and renal failure [4]. In 2022, Heubner and colleagues reported that VET may predict the severity of disease, and might be useful in detecting impaired fibrinolysis in patient with COVID-19-associated ARDS [22], hence there is some rationale to utilize point-of-care VET in daily practice in order to manage CAC appropriately [23]. In 2021, Wang et al. developed a practical TEG-based anticoagulation algorithm to screen and identify patients at high risk for both thrombotic and bleeding complications and to guide their management [24]. Authors suggested that patients with elevated levels of inflammatory biomarkers (C-reactive protein > 30 mg/L, ferritin > 700 ng/mL, lactate dehydrogenase > 300 U/L) and D-dimer (>1000 ng/mL) should be evaluated via TEG. They determined high functional fibrinogen (FF > 32 mm) as the branch point of the algorithm. Patients with high FF and high citrate-kaolin maximum amplitude (CK MA > 69 mm) may be at high risk for thrombotic complications, prompting an escalated full-dose anticoagulant regimen, while screening for arterial and venous thrombosis via ultrasound scan and considering the initiation of antiplatelet therapy. Since COVID-19-associated inflammation is known to increase the level of PAI-1 leading to fibrinolysis shutdown, authors suggested that VET may have a role in guiding the use of rtPA as a potential therapeutic option; however, this working group decided against thrombolysis in this patient population due to bleeding concerns.

Bachler et al. reported that hypercoagulability in ClotPro assays is common in critically ill COVID-19 patients [25]. They found that MCF values were significantly higher in severely ill COVID-19 patients as compared to healthy individuals. Furthermore, CTs on TPA- and IN-test were prolonged significantly in COVID-19 patients, while impaired fibrinolysis indicated by significantly prolonged LT in the TPA assay was also reported (Figure 5). A subgroup analysis of COVID-19 patients with impaired fibrinolytic response vs. patients with normal fibrinolytic activity (cut off LT = 393 s) also showed significantly higher MCFs on every VET assay in the group with prolonged LT, while platelet count and C-reactive protein levels at ICU admission were also significantly higher in this patient population. However, there was no difference in D-dimer levels between the groups. Our results are consistent with these findings: as CAC progressed, CTs and MCFs increased on every assay, while LT prolonged to 755 s.

Medcalf et al. hypothesized that fibrinolysis shutdown may be the result of a consumptive mechanism that exhausts the plasminogen–plasmin pathway, resulting in plasminogen deficiency, and they suggested that fibrinolytic activity can be restored by administration of rtPA and/or its substrate, plasminogen [26]. This hypothesis is supported by the results of a case series from Wang et al. (2020) and research conducted by Della-Morte and colleagues in 2021 [17,27]. Zátroch et al. proposed a simple formula to calculate the appropriate rtPA dose in order to correct tPA levels, and stated that repeated VET measurements are required in order to assess both the efficacy of the treatment and the necessity of subsequent dose adjustments [28]:tPA_L_ = (ML_norm_ − ML_actual_) × 600 ng
V_blood_ = body weight kg × 70 mL
V_plasma_ = V_blood_ × Htc
rtPA = V_plasma_ × tPA_L_
where tPA_L_ is the “lacking” amount of tPA needed to normalize fibrinolysis in a 340 μL of whole blood; ML_norm_ is the normal (100%), and ML_actual_ stands for the measured ML on ClotPro TPA assay; V_blood_ and V_plasma_ are blood and plasma volumes; Htc is hematocrit; rtPA is the dose of alteplase required to restore fibrinolysis.

After substituting the “lacking” tPA, a continuous infusion of rtPA at a dose of 0.05 mg/kg/h used in a previous trial [29] was administered for systemic fibrinolytic treatment (Figure 6).

Despite that there is only a physiological rationale to use this approach in order to substitute the lacking amount of tPA, our findings support this hypothesis. Furthermore, since one size does not fit for all, this personalized approach may give the opportunity to reduce the risk of overtreatment resulting in bleeding complications.

We report a case of a patient with CAC-related PVT and subsequent acute liver failure, treated by a successful VET-guided individualized salvage fibrinolysis. Despite several reports on the existence of COVID-19-associated PVT applying conventional medical treatment (unfractionated heparin, low molecular weight heparin, and oral anticoagulants) [30,31,32,33,34], to our knowledge, this is the first case presentation describing this individualized therapeutic approach.

PVT may result from heritable or acquired disorders of coagulation, hepatocellular cancer, severe intraabdominal hypertension or acute pancreatitis, and it can also develop after abdominal surgeries. It is more common in patients with pre-existing hepatic disease or cirrhosis, and it may be aggravated by the COVID-19-related systemic inflammation [28]. Liver function tests (transaminases, alkaline phosphatase, bilirubin), Doppler-, contrast-enhanced- or endoscopic ultrasound, CT and MRI scans, and investigation of prothrombotic disorders (e.g., antiphospholipid syndrome, Leiden mutation), and coagulation factors (protein C, S, and antithrombin III levels) may support the diagnosis. In terms of differential diagnosis, Budd-Chiari syndrome, cirrhosis, sarcoidosis, porta hepatis masses (regional lymph nodes or cholangiosarcoma causing compression of the portal vein) must be evaluated. Regarding our case, the pathological background (CAC, fibrinolysis shutdown, overwhelming hyperinflammation, and progressing hepatic failure, occlusive thrombus in the portal vein) was evident, hence we excluded other potential causes. As recently reported standard treatment strategies [30,31,32,33,34] have failed, we decided on salvage fibrinolysis after careful consideration of potential risks and benefits. Common indications for the use of rtPA include acute ischemic stroke, myocardial infarction, pulmonary embolism, and deep vein thrombosis; however, there is convincing evidence that selective fibrinolysis via indirect intraarterial (superior mesenteric artery) infusion or directly via the catheter introduced into the portal vein may be useful in very recent non-cirrhotic PVT [35,36]. On the contrary, in general, administration of rtPA is contraindicated when the risk of bleeding or severe complications exceeds the potential benefit. Despite prolonged INR, VET confirmed hypercoagulability and fibrinolysis shutdown, and we excluded any other absolute contraindications for systemic fibrinolytic therapy; therefore, we considered that the treatment can be performed safely and its benefits outweigh the potential risks. Moreover, our VET-guided personalized approach had the potential to decrease the total amount of rtPA, lowering the risk of bleeding complications.

Furthermore, improved oxygenation was considered as a potential additional benefit of fibrinolytic treatment, and this effect was similar to the results reported by Wang and colleagues. However, we cannot conclude whether this improvement resulted from the fibrinolytic therapy or if it was a consequence of other effects.

The main limitation of our case report on a single patient refers to the limited possibility of generalizing the validity of this study and the impossibility of establishing a clear cause-effect relationship. Second, VET-guided individualized thrombolysis has neither clear guidelines regarding the execution of the procedure nor clear evidence regarding the risk-benefit ratio of carrying out such an intervention. Third, we could not measure protein C, protein S, vWF, ADAMTS13 antigen, and antithrombin III levels, which could have contributed more to the assessment of coagulation impairments.

In summary, our observations highlight the importance of close monitoring of hemostasis in critically ill COVID-19 patients in order to reveal and treat the underlying cause of thrombotic complications appropriately.

## 4. Conclusions

Our case presentation highlights the importance of point-of-care viscoelastic testing during the care of severely ill COVID-19 patients, in order to promptly recognize and adequately treat CAC. Conventional coagulation tests are necessary, but their reliability and suitability are limited in this dynamically changing and challenging life-threatening condition. Additionally, we emphasize that in some cases, therapeutic anticoagulant treatment alone may also be insufficient to prevent thrombotic complications in patients suffering from severe SARS-CoV-2 infection. Therefore, we suggest an individualized hemostasis management including extended coagulation laboratory tests and the routine use of VET during the care of critically ill COVID-19 patients. VET-guided personalized fibrinolytic therapy may be a promising tool; however, the exact role and impact of this approach need to be investigated in future trials.

## Figures and Tables

**Figure 1 biomedicines-11-02463-f001:**
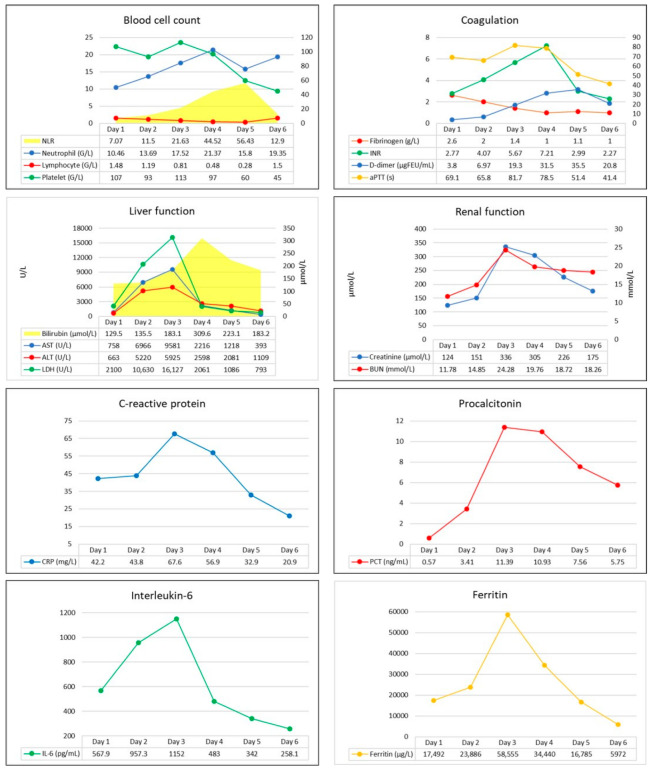
Patient’s laboratory parameters during intensive care. Abbreviations: WBC, white blood cell; NLR, neutrophil to lymphocyte ratio; Plt, platelet count; CRP, C-reactive protein; PCT, procalcitonin; IL-6, interleukin-6; AST, aspartate aminotransferase; ALT, alanine aminotransferase; LDH, lactate dehydrogenase; BUN, blood urea nitrogen; aPTT, activated partial thromboplastin time; INR, international normalized ratio.

**Figure 2 biomedicines-11-02463-f002:**
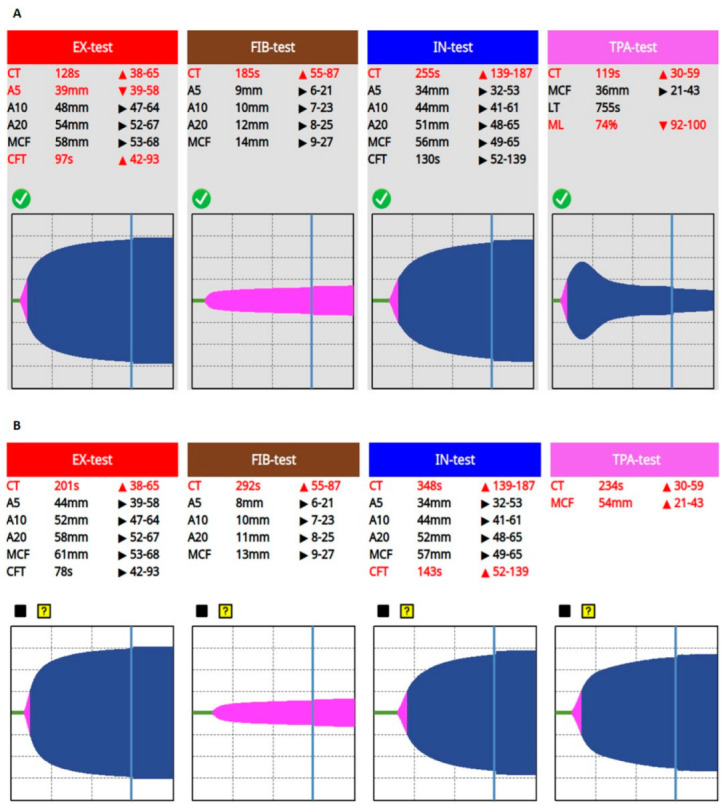
Impairment of fibrinolytic activity: from fibrinolysis resistance to complete fibrinolysis shutdown. After 48 h, fibrinolysis resistance ((**A**), TPA-test, LT = 755 s, ML = 74%) developed, then a few hours later, further progression resulting in fibrinolysis shutdown ((**B**), TPA-test) was confirmed by repeated VET. Abbreviations: CT, clotting time; A5, A10, A20, amplitude; MCF, maximal clot firmness; CFT, clot formation time; LT, lysis time; ML, maximal lysis.

**Figure 3 biomedicines-11-02463-f003:**
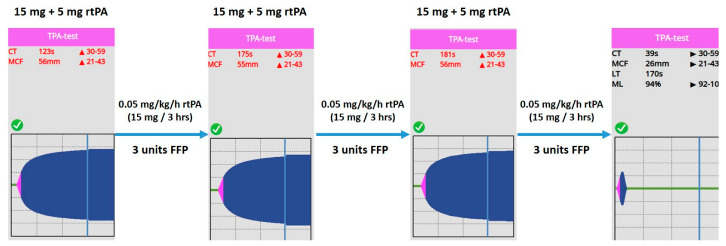
Viscoelastic testing-guided individualized salvage fibrinolytic treatment. Patient’s follow-up viscoelastic profiles obtained during fibrinolytic therapy. Based on patient’s body weight and the results of ClotPro TPA-assays, single doses of 20 mg of rtPA (15 mg “lacking” + 5 mg first hour dose), followed by a continuous infusion (20 mg/ 3 h rtPA) were given. Additionally, fresh frozen plasma (FFP) was administered in order to substitute plasminogen and restore clotting capacity. After three therapeutic cycles—administering a total amount of 105 mg of rtPA and 9 units of FFPs—restoration of fibrinolysis was achieved, as indicated by a normalized LT (170 s). Abbreviations: CT, clotting time; MCF, maximal clot firmness; LT, lysis time; ML, maximal lysis; rtPA, recombined tissue plasminogen activator; FFP, fresh frozen plasma.

**Figure 4 biomedicines-11-02463-f004:**
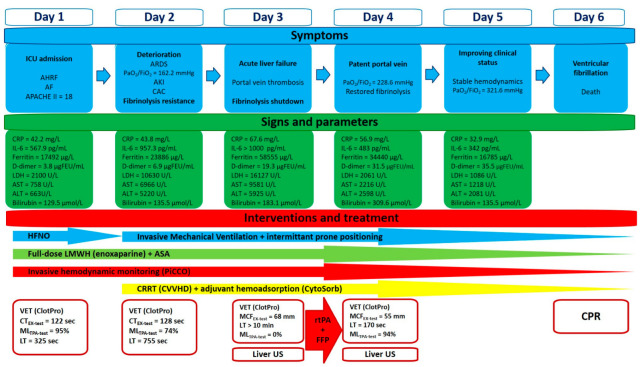
Patient’s clinical course. Signs, symptoms, and assigned interventions and treatment during patient care. Abbreviations: AHRF, acute hypoxemic respiratory failure; AF, atrial fibrillation; CRP, C-reactive protein; IL6, interleukin 6; PaO_2_/FiO_2_, ratio of arterial oxygen partial pressure to fraction of inspired oxygen; ARDS, acute respiratory distress syndrome; AKI, acute kidney injury; CAC, COVID-19-associated coagulopathy; PCT, procalcitonin; AST, aspartate aminotransferase; ALT, alanine aminotransferase; INR, international normalized ratio; HFNO, high flow nasal oxygen; LMWH, low molecular weight heparin; ASA, acetylsalicylic acid; VET, viscoelastic testing; CT, clotting time; LT, lysis time; US, ultrasound; rtPA, recombined tissue plasminogen activator; FFP, fresh frozen plasma; CPR, cardiopulmonary resuscitation.

**Figure 5 biomedicines-11-02463-f005:**
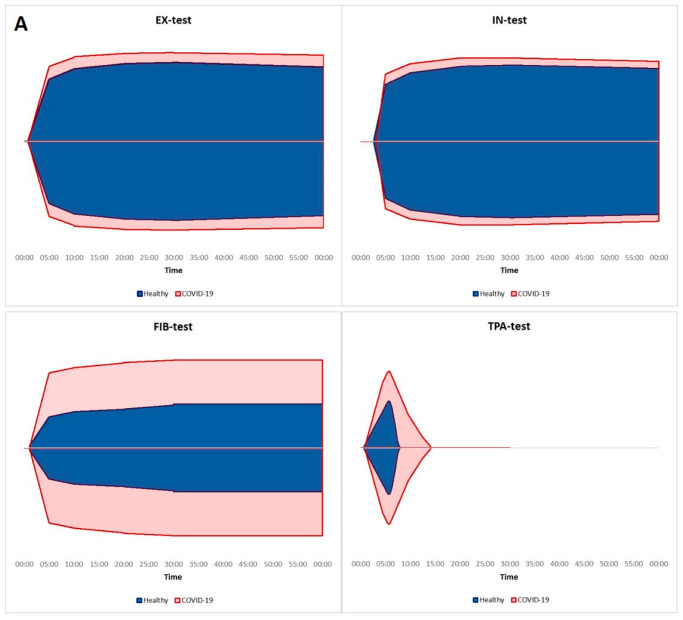
VET profiles of critically ill COVID-19 patients compared to healthy volunteers, based on the results from Bachler et al. [25]. VET assay curves indicated that hypercoagulability is common in patients with severe COVID-19. EX-test MCF of 68 (63–71) mm vs. 61 (58–64) mm (*p* < 0.01), IN-test MCF of 64 (59–69) mm vs. 59 (56–61) mm (*p* < 0.01), and FIB test MCF of 34 (28–39) mm vs. 17 (13–20) mm (*p* < 0.01) were significantly higher in severely ill COVID-19 patients as compared to healthy individuals. Differences between TPA-test CT of 50 (42–90) s vs. 42 (36–46) s (*p* < 0.01), IN-test CT of 188 (168–215) s vs. 159 (153–166) s (*p* < 0.01) were also significant, while impaired fibrinolysis in COVID-19 patients as compared to healthy volunteers was indicated by significantly prolonged LT in the TPA assay: 508 (365–827) s vs. 210 (186–261) s (*p* < 0.01), respectively (**A**). A subgroup analysis of COVID-19 patients with impaired fibrinolytic response with a cut-off value of 393 s of LT vs. patients with normal fibrinolytic activity revealed significantly higher MCFs on every VET assay in the group with fibrinolysis resistance (**B**). EX-test MCF of 60 (57.25–62.75) mm vs. 70.5 (68.25–72) mm, IN-test MCF of 58 (54–59) mm vs. 66.5 (64–69) mm, FIB-test MCF of 26 (24–30) mm vs. 38 (34–40) mm, and TPA-test MCF of 34 (27.5–43.5) mm vs. 62 (55–64.8) mm were found, and all the results were significant (*p* < 0.01). A median LT of 321.5 (278.8–347) s vs. 573 (503–982) s (*p* < 0.01) was found in patients with preserved fibrinolytic activity and patients with fibrinolysis resistance, respectively.

**Figure 6 biomedicines-11-02463-f006:**
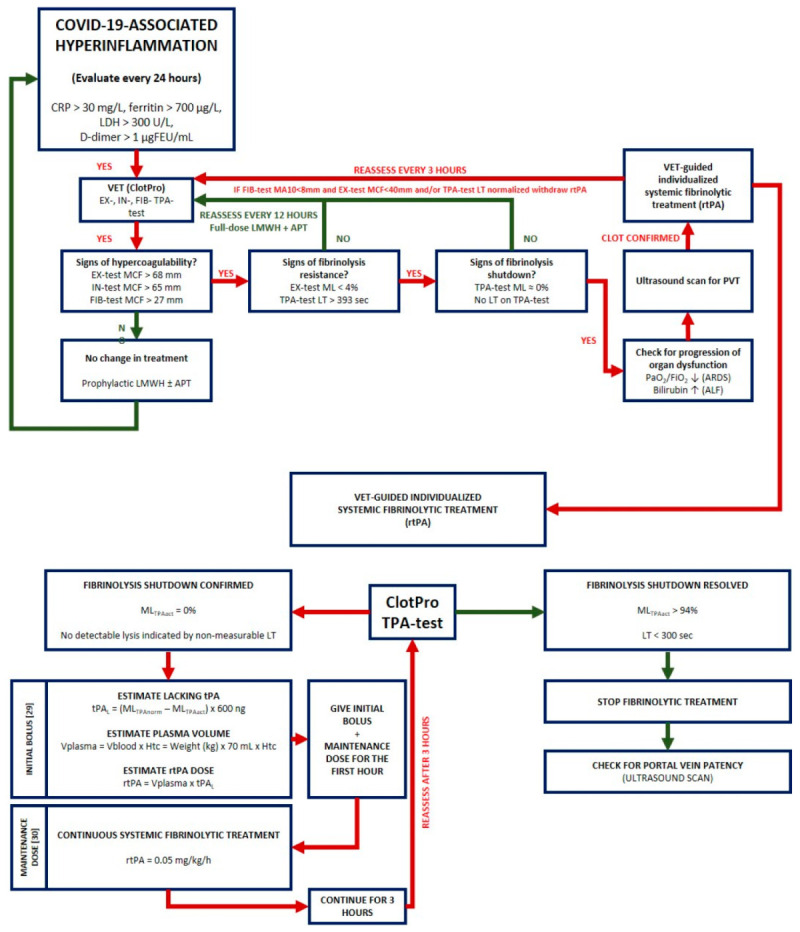
VET-guided therapeutic algorithm for systemic fibrinolysis due to COVID-19-associated portal vein thrombosis. Once a hyperinflammatory state in a severely ill COVID-19 patient is confirmed, VET including EX-, IN-, FIB-, and TPA-test for hypercoagulability is recommended and must be repeated daily. Cut-off values of MCF on the EX-, IN- and FIB-test assays are 68 mm, 65 mm, and 27 mm, respectively. In case of hypercoagulability, signs of fibrinolysis resistance (EX-test ML < 4%, TPA-test LT > 393 s) and fibrinolysis shutdown (no LT on TPA-assay) must be evaluated. In our case, fibrinolysis shutdown and subsequent acute liver failure (ALF) prompted an ultrasound scan to diagnose or exclude portal vein thrombosis (PVT). After PVT was confirmed, we considered a VET-guided individualized systemic fibrinolytic treatment. Initial dose was calculated using the formula by Zátroch et al. [28], while maintenance dose was determined by applying the protocol of a previous study conducted by Hanafy [29]. Abbreviations: CRP, C-reactive protein; LDH, lactate dehydrogenase; VET, viscoelastic testing; MCF, maximal clot firmness; ML, maximal lysis; LT, lysis time; ARDS, acute respiratory distress syndrome; ALF, acute liver failure; PVT, portal vein thrombosis; rtPA, recombined tissue plasminogen activator.

**Table 1 biomedicines-11-02463-t001:** Effects of fibrinolytic therapy on oxygenation and arterial blood gas parameters.

	Before rtPA	24 h after rtPA	48 h after rtPA
PaO_2_/FiO_2_ (mmHg)	162.2	228.6	321.6
PEEP (cmH_2_O)	14	14	12
pH	7.213	7.335	7.421
BE (mmol/L)	−10.6	−5.4	−0.5
Lactate (mmol/L)	11.0	4.2	2.7

Abbreviations: PaO_2_/FiO_2_, ratio of arterial oxygen partial pressure to fraction of inspired oxygen; PEEP, positive end-expiratory pressure; pH, potential of hydrogen; BE, base excess; rtPA, recombined tissue plasminogen activator.

## Data Availability

Deidentified data are available upon request from the corresponding author.

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
