# Peer review of "Thromboelastometry-Guided Individualized Fibrinolytic Treatment for COVID-19-Associated Severe Coagulopathy Complicated by Portal Vein Thrombosis: A Case Report"

_biomedicines, 2023, doi:10.3390/biomedicines11092463_

Round 1

Reviewer 1 Report

Authors reported a case with CPVID-19 associated coagulopathy in monitoring by the thromboelastometry. They found the hypofibrinolysis state in this patient and administrate tPA.

 This manuscript is potentially interesting, several issues arise.

 Title should be reconsidered. I recommend to add some word such as TEG, hypo-fibrinolysis or tPA.

 Abstract should be clear.

 Was an antithrombin, protein C, protein S or antiphospholipid antibody examined in this patient?

 Was the HIT antibody examined in this patient?

 Authors should show the clinical course in Figure.

 When was the t-PA administrated after the onset of thrombosis? In acute myocardial infarction or acute cerebral infarction, t-PA should be immediately administrate after the onset.

 Authors should explain the relationship between hemostatic abnormality and TEG.

 Was the TEG measured before administration of heparin.

  Authors should show the criteria for treatment with t-PA.

Author Response

Dear Reviewer,

I am grateful for your accurate work and constructive critical remarks and suggestions on our paper. I hope our revised manuscript will fulfill the requirements for publication. According to your suggestions and comments we made efforts to correct all errors and issues arised during the review process. Please also see below our point-by-point responses to the comments.

We would like to resubmit the revised version for your kind consideration. We hope, these changes make our manuscript more valuable and worth for publication.

Point 1: Title should be reconsidered. I recommend to add some word such as TEG, hypo-fibrinolysis or tPA.

Response 1: Thank you for this recommendation. We changed the title of our manuscript to "Thromboelastometry guided individualized fibrinolytic treatment for COVID-19 associated severe coagulopathy complicated by portal vein thrombosis: A case report". We believe, this title is much more relevant than the earlier one.

Point 2: Abstract should be clear.

Response 2: Thank you for this remark. We made efforts to make abstract more clear and understanable.

Point 3: Was an antithrombin, protein C, protein S or antiphospholipid antibody examined in this patient?

Response 3: Thank you for this valuable question. Unfortunately we could not measure these coagulation parameters.

Point 4: Was the HIT antibody examined in this patient?

Response 4: Unfortunately we could not examine HIT antibody.

Point 5: Authors should show the clinical course in Figure.

Response 5: Thank you for this recommendation. We replaced Table 1 with a figure. It makes much more understanable and followable the clincal course of our patient, and adds value to our manuscript.

Point 6: When was the t-PA administrated after the onset of thrombosis? In acute myocardial infarction or acute cerebral infarction, t-PA should be immediately administrate after the onset.

Response 6: Thank you for your question. TPA was administered only after the portal vein thrombosis was revealed and contraindications were excluded. Since portal vein thrombosis is not an acute emergency, we had some time for consideration, however we decided to perform fibrinolytic therapy due to the progression of liver failure and deterioration of our patient.

Point 7: Authors should explain the relationship between hemostatic abnormality and TEG.

Response 7: Thank you for this suggestion. We hope, these sentences are eligible. "Prolonged CTs, normal clot formation times (CFT) and mean clot firmness (MCF) were seen on every VET assay (IN-test, EX-test, FIB-test and TPA-test), while TPA-test indicated decreased fibrinolytic activity, then a few hours later a complete fibrinolysis shutdown was confirmed. Since prolonged CT on VET assay may be a result of ongoing anticoagulation therapy, or it can be caused by the low levels of clotting factors, 600 IU of prothrombin complex concentrate was given in order to normalize coagulation."

Point 8: Was the TEG measured before administration of heparin.

Response 8: First VET measurement was performed only after therapeutic anticoagulation applying enoxaparine. Despite, in general, LMWH does not alter CT on VET EX- an IN-test assays, there is convincing evidence, that this is not completley true, especially enoxaparine when a high therapeutic dose is administered. 

Point 9: Authors should show the criteria for treatment with t-PA.

Response 9: Thank you for this important remark. We completed the discussion section with the following paragraph: "Common indications for the use of rtPA include acute ischaemic stroke, myocardial infarction, pulmonary embolism and deep vein thrombosis, however there is convincing evidence that selective fibrinolysis via indirect intraarterial (superior mesenteric artery) infusion or directly via the catheter introduced into the portal vein may be useful in very recent non-cirrhotic PVT [30,31]. On the contrary, in general, administration of rtPA is contraindicated when the risk of bleeding or severe complications exceed the potential benefit. Despite prolonged INR, VET confirmed hypercoagulability and fibrinolysis shutdown, and we excluded any other absolute contraindications for systemic fibrinolytic therapy, therefore we considered that the treatment can be performed safely and its benefits outweigh the potential risks."

Reviewer 2 Report

The authors present a well-structured case report, providing a detailed description of the patient's condition. Moreover, it is well-written and easy to follow. However, I noticed some minor problems that should be addressed before publication.

1. The article could be improved by including in the discussion section the limitations and potential risks associated with the treatment used in this case.

2. It would be beneficial to include a more detailed explanation of viscoelastic thromboelastometry testing (VET) in the introduction section.

3. In terms of language and writing style, the article is generally well-written and concise. However, some sentences could be rephrased or clarified to improve readability and flow (introduction and discussion).

4. It would be beneficial to include in the discussion section a brief mention of the differential diagnosis in this case, along with an explanation of the key factors that supported the diagnosis of COVID-19-associated severe coagulopathy in this specific case.

5. Since COVID-19 is strongly associated with thrombotic complications, it would be valuable to expand the case report by a following reference: doi: 10.1016/j.thromres.2023.01.016. This addition would enrich the discussion on thrombotic mechanisms, especially in hospitalized patients.

6. Please summarized in a small paragraph (discussion) current therapeutic approach in existing reports on COVID-19 associated PVT, and, highlight the utility of your approach.

7. Please create a Figure in the Introduction section highlighting the key clinical information related to COVID-19-associated coagulopathy in a visually striking and clinically relevant manner.

8. The the last sentence in Conclusions should be completely rephrased to strengthen the impact and grab the reader's attention.

Author Response

Dear Reviewer,

I am grateful for your accurate work and constructive critical remarks and suggestions on our paper. I hope our revised manuscript will fulfill the requirements for publication. According to your suggestions and comments we made efforts to correct all errors and issues arised during the review process. Please also see below our point-by-point responses to the comments.

We would like to resubmit the revised version for your kind consideration. We hope, these changes make our manuscript more valuable and worth for publication.

Point 1: The article could be improved by including in the discussion section the limitations and potential risks associated with the treatment used in this case.

Response 1: Thank you for this important suggestion. We completed the discussion section with the following: "On the contrary, in general, administration of rtPA is contraindicated when the risk of bleeding or severe complications exceed the potential benefit. Despite prolonged INR, VET confirmed hypercoagulability and fibrinolysis shutdown, and we excluded any other absolute contraindications for systemic fibrinolytic therapy, therefore we considered that the treatment can be performed safely and its benefits outweigh the potential risks. Moreover, our VET guided personalized approach had the potential to decrease the total amount of rtPA lowering the risk of bleeding complications. (...) The main limitation of our case report on a single patient refer to the limited possibility of generalizing the validity of the study and the impossibility of establishing a clear cause-effect relationship. Second, VET guided individualized thrombolysis has neither clear guidelines regarding the execution of the procedure nor clear evidence regarding the risk-benefit ratio of carrying out such an intervention. Third, we could not measure protein C, protein S, vWF, ADAMTS13 antigen and antithrombin III levels, which could have contributed more to the assessment of coagulation impairments."

Point 2: It would be beneficial to include a more detailed explanation of viscoelastic thromboelastometry testing (VET) in the introduction section.

Response 2: Thank you for this remark. We replaced the detailed explanation of VET from a figure legend into the introduction section: "The most commonly used VET assays are the EX-, IN- and FIB-tests. EX-test and IN-test assays model the extrinsic and intrinsic coagulation pathways respectively, while the FIB-test containing platelet inhibitors (cytochalasin D and a synthetic GP2b3a antagonist) represents the tissue factor induced activation of coagulation and the strength of fibrin polymerisation. Viscoelastic traces can be divided into two parts. The first phase represents clotting (initiation, amplification, clot and fibrin formation), while the second phase provides information about fibrinolysis. In details, clotting time (CT) refers to the time in seconds needed for the first detectable clot formation; clot formation time (CFT) indicates the achievement of clot firmness in seconds; amplitude 5, 10 and 20 (A5, A10, A20) are the early parameters of clot quality measured at 5, 10 and 20 minutes timepoints after initiation, respectively; and maximal clot firmness (MCF) represents the maximal strength of the clot. Maximal lysis (ML) expressed as a percentage at the end of the measurement is a marker of fibrinolytic activity. It is an appropriate parameter in case of hyperfibrinolysis, but it only seems to be a surrogate marker when hypofibrinolysis is suspected. In summary, VET gives clear information mainly about the coagulation part of the hemostatic system, however, the novel ClotPro device (enicor GmbH, Munich, Germany; Haemonetics Corporation) has also the potential to evaluate the fibrinolysis shutdown phenomenon via the TPA assay containing 600 ng of recombinant tissue plasminogen activator (rtPA) that is added to 340 µL of whole citrated blood will result in complete fibrinolysis indicated by normal lysis time (LT < 300 s), and 100% ML, if the fibrinolytic activity is preserved. In contrast, both prolonged LT, and / or incomplete lysis (ML < 100%) indicate fibrinolysis resistance."

Point 3: In terms of language and writing style, the article is generally well-written and concise. However, some sentences could be rephrased or clarified to improve readability and flow (introduction and discussion).

Response 3: Thank you for your feedback. We made efforts to rephrase some sentences those were thought complicated or difficult to understand.

Point 4: It would be beneficial to include in the discussion section a brief mention of the differential diagnosis in this case, along with an explanation of the key factors that supported the diagnosis of COVID-19-associated severe coagulopathy in this specific case.

Response 4: Thank you for this important issue. We inserted a brief paragraph on differential diagnostic steps and supporting factors. "PVT may result from heritable or acquired disorders of coagulation, hepatocellular cancer, severe intraabdominal hypertension or acute pancreatitis, and it can also develop after abdominal surgeries. It is more common in patients with pre-existing hepatic disease or cirrhosis, and it may be aggravated by the COVID-19 related systemic inflammation [29]. Liver function tests (transaminases, alkaline phosphatase, bilirubin), Doppler-, contrast-enhanced- or endoscopic ultrasound, CT and MRI scans and investigation of prothrombotic disorders (eg.: antiphospholipid syndrome, Leiden mutation) and coagulation factors (protein C, S, and antithrombin III levels) may support the diagnosis. In terms of differential diagnosis Budd-Chiari syndrome, cirrhosis, sarcoidosis, porta hepatis masses (regional lymph nodes or cholangiosarcoma causing compression of the portal vein) must be evaluated. Regarding our case, the pathological background (CAC, fibrinolysis shutdown, overwhelming hyperinflammation and progressing hepatic failure, occlusive thrombus in the portal vein) was evident, hence we excluded other potential causes."

Point 5: Since COVID-19 is strongly associated with thrombotic complications, it would be valuable to expand the case report by a following reference: doi: 10.1016/j.thromres.2023.01.016. This addition would enrich the discussion on thrombotic mechanisms, especially in hospitalized patients.

Response 5: Thank you for this suggestion. We added the key message of the suggested publication from Wójcik et al to the discussion section.

Point 6: Please summarized in a small paragraph (discussion) current therapeutic approach in existing reports on COVID-19 associated PVT, and, highlight the utility of your approach.

Response 6: Please see our answer in A4.

Point 7: Please create a Figure in the Introduction section highlighting the key clinical information related to COVID-19-associated coagulopathy in a visually striking and clinically relevant manner.

Response 7: We created a figure about the pathomechanism of COVID-19 associated coagulopathy. We believe this figure is easy to interpret and added value to our manuscript.

Point 8: The the last sentence in Conclusions should be completely rephrased to strengthen the impact and grab the reader's attention.

Response 8: Thank you for drawing attention to this important issue. We rephrased the last sentences in the conclusion section to highlight the main message and the lessons should be learned from our case. "Additionally, we emphasize, that in some cases, therapeutic anticoagulant treatment alone may also be insufficient to prevent thrombotic complications in patients suffering from severe SARS-CoV-2 infection. Therefore, we suggest an individualized hemostasis management including extended coagulation laboratory tests and the routine use of VET during the care of critically ill COVID-19 patients. VET guided personalized fibrinolytic therapy may be a promising tool, however the exact role and impact of this approach need to be investigated in future trials."

Round 2

Reviewer 1 Report

Although revised manuscript has been partially improved, several issues arise.

1)    Abstract is still not clear.

2)    Figure for clinical course should contain symptom and treatment.

3)    Normal range in healthy volunteers and values in COVID-19 patients without thrombosis of TEG may be helpful.

4)    There are many figures. These are required to combine or delete.

5)    Figure 1 may not be necessary in case report.

6)    Algorism for diagnosis and treatment of thrombosis in COVID-19 patients using TEG may be helpful

Author Response

Dear Reviewer,

I would like to thank you again for your accurate work and constructive critical remarks and suggestions on our paper. I hope our revised manuscript will fulfill the requirements for publication. According to your suggestions and comments we made efforts to correct all errors and issues arised during the review process. Please also see below our point-by-point responses to the comments.

We would like to resubmit the revised version for your kind consideration. We hope, these changes make our manuscript more valuable and worth for publication.

Point 1: Abstract is still not clear.

Response 1: We entirely reconstructed the Abstract. We hope, that all relevant information are included, and the abstract became clear and consistent.

Point 2: Figure for clinical course should contain symptom and treatment.

Response 2: Thank you for explanation. Unfortunately we misunderstood your request during review round 1. We constructed a new figure (Figure 4) including patient's clinical course, signs, symptoms and assigned interventions and treatment.

Point 3: Normal range in healthy volunteers and values in COVID-19 patients without thrombosis of TEG may be helpful.

Response 3: Thank you for this important remark. To improve our manuscript we referenced a publication from Bachler et al. (Bachler, M; Bösch, J; Stürzel, DP; et al. Impaired fibrinolysis in critically ill COVID-19 patients. Brit. J. Anaesth. 2021, 126, 590-598.) comparing common VET assay findings of healthy volunteers with the results of COVID-19 patients. Furthermore this publication includes a subgroup analysis comparing COVID-19 patients with and without fibrinolysis resistance with a cut off value of 393 s of LT on TPA assay. 

Point 4: There are many figures. These are required to combine or delete.

Response 4: Thank you for this suggestion. We reduced the number of figures. Finally, our manuscript includes 4 figures and 1 table.

Point 5: Figure 1 may not be necessary in case report.

Response 5: Thank you for this suggestion. However, we had some concerns about this issue. We constructed this figure (Figure 1), because it was suggested during review round 1 by another reviewer. However, taking your advice we deleted it. Furthermore, we think that the main pathophysiological background of CAC is well described in Discussion section. 

Point 6: Algorism for diagnosis and treatment of thrombosis in COVID-19 patients using TEG may be helpful

Response 6: Thank you for this important advice. However, we do not think we can establish a clear algorithm based on a case report of a single patient. Despite there is convincing evidence about the value and usefulness of TEG / TEM in some clinical scenarios (eg.: cardiac surgery), the exact role of VET during the care of critically ill COVID-19 patients is still unclear. However, we referenced a possible and promising TEG based algorithm developed and reported by Wang and colleagues in 2021 (Wang, J; Hajizadeh, N; Shore-Lesserson, L. The Value of Thromboelastography (TEG) in COVID-19 Critical Illness as Illustrated by a Case Series. J. Cardiothorac. Vasc. Anesth. 2022, 36, 2536-2543.). This suggested practical TEG based anticoagulation algorithm may serve to screen and identify patients at high risk for both thrombotic and bleeding complications and to guide their management.

Round 3

Reviewer 1 Report

Revised manuscript has not been sufficiently improved.

Author Response

Dear Reviewer,

I would like to thank you again for comments and suggestions on our paper. I hope our revised manuscript improved significantly and will fulfill the requirements for publication. According to your suggestions and comments we made efforts to correct all issues arised during the review process. Changed text and paragraphs are highlighted with yellow background. Please also see below our point-by-point responses to your comments.

We would like to resubmit the revised version for your kind consideration. We hope, these changes make our manuscript more valuable and worth for publication.

Point 1: Abstract is still not clear. Authors should show how to guide
using TEG. What is the parameter? What is the adequate cutoff value?

Response 1: Thank you for this comment. We reconstructed the abstract, and entered the requested parameters. During the treatment of PVT caused by fibrinolysis shutdown, the main VET paramter was Lysis time (LT). Cut off value of LT for fibrinolysis resistance was 300 sec (Coupland, LA; Rabbolini, DJ; Schoenecker, JG; et al. Point-of-care diagnosis and monitoring of fibrinolysis resistance in the critically ill: results from a feasibility study. Crit. Care 2023, 27, 55.) and panic value was 393 sec (Bachler, M; Bösch, J; Stürzel, DP; Hell, T; Giebl, A; Ströhle, M; Klein, SJ; Schäfer, V; Lehner, GF; Joannidis, M; Thomé, C; Fries, D. Impaired fibrinolysis in critically ill COVID-19 patients. Brit. J. Anaesth. 2021, 126, 590-598.)

Point 2: Figure for clinical course should contain symptom, treatment,
laboratory data and parameter of guide.

Response 2: Thank you for explanation. We reconstructed this figure, and entered all important and relevant data.

Point 3: Normal range in healthy volunteers and values in COVID-19
patients without thrombosis of TEG may be helpful. This value in figure
may be helpful.

Response 3: Thank you for this valuable suggestion. Based on the results from a trial conducted by Bachler et al. (Brit. J. Anaesth. 2021, 126, 590-598.) we created a figure with hypothetical VET curves indicating the differences between the results of healthy volunteers and critically ill COVID-19 patients.

Point 4: Algorism for diagnosis and treatment of thrombosis in COVID-19
patients using TEG may be helpful. How do author monitor fibrinolytic
state using parameter? What is the cutoff value? What is panic value?

Response 4: Thank you for this suggestion. As we stated in an earlier review round, we do not think we can establish a generally valuable algorithm based on a case report of a single patient. However, we created a figure including our case-based therapeutic algorithm, that we used during patient care. We hope, this figure gives clear information about our therapeutic decisions.

Round 4

Reviewer 1 Report

Revised manuscript has been improved. I have no further comment.